# Advanced Spheroid, Tumouroid and 3D Bioprinted In-Vitro Models of Adult and Paediatric Glioblastoma

**DOI:** 10.3390/ijms22062962

**Published:** 2021-03-15

**Authors:** Louise Orcheston-Findlay, Samuel Bax, Robert Utama, Martin Engel, Dinisha Govender, Geraldine O’Neill

**Affiliations:** 1Children’s Cancer Research Unit, The Children’s Hospital at Westmead, Sydney, NSW 2145, Australia; Louise.OrchestonFindlay@health.nsw.gov.au (L.O.-F.); Samuel.Bax@health.nsw.gov.au (S.B.); 2Inventia Life Science Pty Ltd., Sydney, NSW 2015, Australia; Robert.Utama@inventia.life (R.U.); Martin.Engel@inventia.life (M.E.); 3Cancer Centre for Children, The Children’s Hospital at Westmead, Sydney, NSW 2145, Australia; Dinisha.Govender@health.nsw.gov.au; 4Children’s Hospital at Westmead Clinical School, Faculty of Medicine and Health, University of Sydney, Sydney, NSW 2145, Australia; 5School of Medical Sciences, Faculty of Medicine and Health, University of Sydney, Sydney, NSW 2145, Australia

**Keywords:** glioma, model, microenvironment, spheroid, tumouroid, organoid, bioprinting, microfluidics

## Abstract

The life expectancy of patients with high-grade glioma (HGG) has not improved in decades. One of the crucial tools to enable future improvement is advanced models that faithfully recapitulate the tumour microenvironment; they can be used for high-throughput screening that in future may enable accurate personalised drug screens. Currently, advanced models are crucial for identifying and understanding potential new targets, assessing new chemotherapeutic compounds or other treatment modalities. Recently, various methodologies have come into use that have allowed the validation of complex models—namely, spheroids, tumouroids, hydrogel-embedded cultures (matrix-supported) and advanced bioengineered cultures assembled with bioprinting and microfluidics. This review is designed to present the state of advanced models of HGG, whilst focusing as much as is possible on the paediatric form of the disease. The reality remains, however, that paediatric HGG (pHGG) models are years behind those of adult HGG. Our goal is to bring this to light in the hope that pGBM models can be improved upon.

## 1. Introduction

Patients with high-grade glioma (HGG) brain tumours, including adult glioblastoma (GBM) and paediatric gliomas including paediatric GBM (pGBM) and diffuse midline glioma (DMG) have a five-year survival rate of 20% [1] and 5% [2], respectively. The treatment regime for adults is surgery, radiotherapy and chemotherapy, to which less than half of patients respond and the remainder gain only a few additional months of life [1,3]. There is no standard chemotherapy regimen designed for paediatric HGG (pHGG) and patients are given radiotherapy and/or surgery on a case-by-case basis [4,5,6,7,8]—there is no evidence that Temozolomide is beneficial [7,9,10].

The fate of HGG patients has improved only marginally in the last 40 years [6,10] and given intrinsic drug resistance and difficulty or impossibility of removal [5,11,12,13], there is an urgent need for effective therapies.

High-grade glioma cells exhibit both individual and collective modes of migration, influenced by cell–cell connections between neighbouring glioma cells and neural cells, and by the physico-chemical features of the surrounding brain tissue [14,15,16,17,18]. Previous models of diffuse intrinsic pontine glioma (DIPG) (recently reclassified as a subgroup of DMG) have suggested that cells disseminate throughout the pons both individually and collectively [19], although this study investigated adult GBM lines injected into the pons and the implications for DMG specifically therefore remain to be confirmed. More recently, cooperativity between different DMG subclones resulted in enhanced invasive capacity of otherwise poorly invasive DMG subclones [20]. Furthermore, the inter- and intratumoural heterogeneity is extreme [21,22], resulting in inaccurate or misleading predictions of clinical response from simplified laboratory models—wasting precious time and resources.

In this review, an introduction to crucial aspects for modelling HGGs in adults and children will be given. This will be followed by a comparison of several methods for the generation of sophisticated preclinical models that incorporate the required attributes in a controlled manner. Notably, modelling of adult HGG has progressed further than pHGG, thus where possible we extrapolate from models of HGG and consider how they might be applied in studies of pHGG.

## 2. The Tumour Microenvironment

The location of GBM is a significant barrier to successful treatment. As well as making surgical resection difficult, the extracellular matrix (ECM) composition [23,24], tissue mechanics [15,25] and stromal-cell interactions [26,27,28] within the brain elicit unique tumour qualities. The brain provides tissue niches for stem-like cancer stem cells (CSCs) and hypoxic regions develop, which further enables drug resistance and recurrence [29,30]. Each of these will be briefly introduced before progressing to discussions of their controlled incorporation into disease models.

### 2.1. Glioma Stem Cells

The population of radially migrating cells at the periphery of HGGs commonly host CSCs—also termed glioma stem cells (GSCs) and glioma initiating cells (GICs) [31,32,33,34,35]—that can self-renew [36], are drug resistant and radioresistant [27,32,34,37] and indicate poor prognosis [11,38]. Studies suggest that CSCs are also present in DMG [35]. They are thought to be a key contributor to high recurrence rates and therapy resistance [30,36,39].

### 2.2. Hypoxia

Dense and fast-growing regions of GBM host sharp solute gradients [40]. As a result, hypoxia forms, which increases drug resistance [41]. It appears that hypoxia is also a factor in both pHGG and DMG [42,43]. Furthermore, via the increased expression of hypoxia inducible factors (HIFs), the stem-like functions of CSCs are thought to be enhanced [44]. The hypoxic landscape is heterogeneous between tumours and patients [40], so modelling this in a patient-specific manner is not trivial.

### 2.3. Extracellular Matrix

The ECM composition of healthy brain is almost void of fibrous proteins seen in high concentrations in other parts of the body [44,45]. It is instead rich in glycosaminoglycans (GAGs), among these, hyaluronic acid (HA)—a GAG that regulates brain tissue mechanics, organisation and signalling [44,46]—is present in brain ECM in high concentrations [15,38]. In GBMs, GAGs are overexpressed compared to normal brain three- to four-fold and are associated with increased proliferation, invasion, resistance, recurrence and poor prognosis [47]. Fibrillar collagens are upregulated in subsets of gliomas, thus potentially influence glioma behaviour within the tumour mass [15]. Cells can disseminate along the basement membrane lining blood vessels in the brain, rich in laminins and collagen IV [14]. Thus, it is essential to incorporate or mimic these ECM components as part of pre-clinical models.

### 2.4. Tumour Interactions with Non-Tumour Cells

The tumour stroma may provide a supportive niche for CSCs, driving invasion and recurrence [26], however the inflammatory microenvironment of pHGG is reported to be fundamentally different to that of adult HGG. Reports suggest the presence of a stem cell population present in adjacent healthy stroma, termed glioma associated mesenchymal stem cells (GA-MSCs), has been shown to excrete exosomes that increase the proliferation of CSCs. Furthermore, CSCs pre-treated with exosomes derived from GA-MSCs decreased the survival of the mice in which they were implanted [26]. Glioblastoma cells are also known to induce co-expression with surrounding GA-MSCs [12] and recruit and polarise tumour-associated macrophages to aid in their continued proliferation and survival [48].

Interactions between tumour cells and healthy neuronal cells also play a role [49]. Increased neural activity in pGBM patient-derived xenograft (PDX) models resulted in increased tumour proliferation [49]. Moreover, GBMs contain sub-populations that resemble oligodendroglial precursor cells (OPCs) that express numerous synaptic genes and have been observed to form synapses with healthy neurons [49]. This activity is thought to support progression [28,49] and highlights the need to consider the contribution of healthy brain tissue in models of HGG.

### 2.5. Tumour Microtubules

In recent years, tumour microtubules (TMs) have been suggested to have an important role in GBM invasion and survival [29]. Their presence is highly influenced by tumour type and grade and is significantly correlated with poor prognosis [16,30]. Additionally, it appears that DMG may similarly elaborate tumour microtube connections [49]. Tumour microtubules interconnect the cytoplasm of cells through gap junctions and are responsible for two distinct functionalities [16,29,30]; (i) to probe the environment at the leading edge of the tumour and subsequently direct invasion and (ii) to provide connections between a proportion of single glioma cells [16,29]. Together, this results in a large network of functional and resistant glioma cells that is able to repair damaged cells within and around the network [16,29]. Several mechanisms of therapy resistance linked to these systems have been suggested [16,29]. For example, through the distribution—and subsequent dilution—of toxicity and the replacement of damaged cell components.

### 2.6. Mechanical Properties

The brain is viscoelastic and is reported to have a particularly low physical stiffness of around 1–10 kPa [50] that varies depending on the brain region and over micron-scale distances [46,51]. Glioblastomas are most frequently seen to arise in the frontal and temporal lobes of the cerebral hemispheres, while DMG comprises tumours in the midbrain and within as well as outside the brainstem. The brainstem has been reported to be the stiffest of the brain regions [52] and thus in future it will be important to consider this parameter in models of brain cancer, matching the tumours to the biomechanical features of their anatomical location. It has now been established that through the secretion and organisation of a dense fibrillar matrix, many solid tumours become increasingly stiff, resulting in tumour progression [53]. Thus, it is important to establish whether HGG similarly display a rigid matrix relative to the surrounding soft tissue. Current reports of the mechanical properties of GBMs suggest that the tumours are alternatively either stiffer or softer than surrounding healthy brain [44,46,54]. These opposite results may stem from differing measurement techniques. Most measurements are performed on ex-vivo brains, contrasting more recent applications of magnetic resonance elastography (MRE) (see Box 1) where imaging is performed on intact brains in-vivo. A significant limitation of all analyses of the mechanical properties of tissues is that the readout depends on the precise parameters (application and duration of force, for example) and technical approach used. Thus, to generate models that appropriately mimic the biomechanical features of the relevant brain tumour microenvironment (TME), approaches are required that allow direct comparison of the mechanical attributes of the model with tissue measurements. Recent successful adaptations of MRE analyses to small organoid cultures provide a potential avenue for this [55].

Box 1Magnetic Resonance Elastography.Recently, an established technique has been re-purposed to non-invasively characterise the viscoelastic properties of healthy brain [56] and its malignancies [57]. This technique, termed MRE, involves applying shear waves to the tissue of interest and observing their attenuation through the sample with magnetic resonance imaging (MRI), from which mechanical properties are extracted. This technique is traditionally applied clinically to stage liver fibrosis [58], but has more recently been validated for measuring mechanical properties of brain cancers [57]. The technique yields the storage modulus (G${}^{\prime }$), loss modulus ($G^{\prime \prime }$) and phase angle (ϕ) for the tissue of interest—measures of shear stiffness, shear viscosity and liquid/solid characteristics, respectively. The stiffness represents the deformation with applied shear force, the viscosity signifies the deformation over time while under a constant force and the phase angle denotes behaviour in the range from elastic solid to viscous liquid [54]. These are related via $Tan\left(\phi \right)=G^{\prime }/G^{\prime \prime }$ and the complex modulus $|G^{*}|$, representing resistance to deformation, is equated to the stiffness and viscosity via $|G^{*}|=\sqrt{G^{\prime \prime 2}+G^{\prime 2}}$ [57].Multifrequency MRE (MMRE) has been used to effectively probe the full range of viscoelastic behaviour in anisotropic brain tissues [55,59]. In GBMs, a wide range of $|G^{*}|$ values has been observed within and between tumours; broadly, there is an increased loss of stiffness the higher the grade and GBM is the softest of all primary brain tumours [60,61,62]. The complex modulus can be used to distinguish between GBM and normal-appearing white matter (NAWM) with a high degree of certainty [63], but such a large difference in $|G^{*}|$ is not always the case, as the data presented in Table 1 demonstrate. The phase angle, as well as a possible diagnostic factor, may also provide insight into the invasion of GBM into stiffer NAWM. Viscous fingering—a physical phenomenon where softer material is able to passively infiltrate stiffer material—has been suggested, rather than active tissue displacement by more rigid tumours [54]. Furthermore, the intertumoural heterogeneity of GBM has been demonstrated by a stiffness variance of more than 20% [64] and five out of 22 participants exhibited a GBM tumour that is stiffer than surrounding NAWM as measured by MRE [65]. It remains to be seen if this is a patient-specific trait.

**Table 1 ijms-22-02962-t001:** The complex modulus $|G^{*}|$ and phase angle ϕ inhuman GBM and NAWM measured using MRE, demonstrating its pre-operative diagnostic capabilities. Entries are presented inreverse dateorder.

Sample Size, Age Range (y)	Range of Excitation Frequency (Hz)	Complex Modulus $|G^{*}|$ (rad)			Phase Angle ϕ			Ref.
		GBM	NAWM	$\frac{GBM}{NAWM}$	GBM	NAWM	$\frac{GBM}{NAWM}$	
9, 60–80	30–60	1.10 ± 0.29	1.81 ± 0.23	0.65 ± 0.04	0.62 ± 0.19	0.36 ± 0.10	0.54 ± 0.15	[54]
6, 25–68	60	1.7 ± 0.5	3.3 ± 0.7	—	—	—	—	[63]
11, 42–86	30–60	1.37 ± 0.26	1.64 ± 0.21	0.64 ± 0.10	0.85 ± 0.22	0.44 ± 0.07	0.70 ± 0.11	[61]
22, 18–86	30–60	1.32 ± 0.26	1.54 ± 0.27	0.58 ± 0.07	0.88 ± 0.19	0.37 ± 0.08	0.66 ± 0.15	[65]
3, 53–69	45	1.24 ± 0.31	2.11 ± 0.31	0.41 ± 0.06	0.59 ± 0.09	0.30 ± 0.04	0.74 ± 0.19	[60]

Accurate measurement of mechanical properties of HGGs and healthy brain tissue is crucial since it is a determinant of cell behaviour, including drug response [66], important for understanding HGG and other cancers as well as for guiding the production of representative models.

## 3. Traditional In-Vitro Models of High Grade Glioma

While much research has been undertaken using two-dimensional (2D) culture, it is increasingly appreciated that such models are ill equipped to reproduce the multifaceted characteristics of HGGs. Moreover, the use of continuously cultured cell lines that have been maintained in the presence of serum has been shown to significantly alter the cellular phenotype and genotype to the extent that they retain few features of the primary tumour [67]. Increasingly, studies employ patient-derived cell lines that are maintained in serum-free conditions, in neural-cell specific medium and at low passage to retain phenotypes and genotypes [68]. Thus, in this review we do not discuss analyses in standard 2D culture and focus on studies using patient-derived cell lines grown under serum-free conditions.

Mouse orthotopic xenografts of patient-derived cells display genetic heterogeneity [24] and patient-specific drug responses [69] and responses to fluid shear [70] and thus provide disease relevance. Pepin et al. found that *IDH1*-mutated PDX GBM models were significantly softer than wild-type GBM via MRE [63]. The reason for this—reducing tenascin C expression—was also discovered using mouse PDX models [71].

The caveats of mouse models are that tumours can take months to establish [72] and mice are commonly immunocompromised [73]. Humanised [25] and AVATAR mouse models [44] can address some of these issues, but ethical considerations prevent the use of animal models for high-throughput drug screening. There are published works that utilise mouse models for pGBM [37,69] and DMG [13,74] research, but for reasons stated above, the use of such mouse models is not the best solution for some applications. Alternative 3D modalities are becoming more sophisticated and may soon limit the need for animal models. These 3D models are the focus of the remainder of this article.

### 3.1. Three-Dimensional In-Vitro Models of High Grade Glioma

With the rapid development of new 3D culture methods, there is some confusion around the nomenclature used to describe different resulting cultures. The terms tumouroid and organoid are often used interchangeably, however this creates confusion when we then start to consider the combination of tumouroids (3D spheroid cultures derived directly from patient tumours) together with organoids (organ-like 3D cultures often derived from pluripotent stem cells) in order to provide an in-vivo-like tissue microenvironment for the tumour cells. Both tumouroids and organoids are 3D structures, composed of distinct cell types and grown under conditions to promote stem cell maintenance. However, while all tumouroids consist of malignant cells—with the exception of organoids, in which cancer-promoting mutations have been introduced [75,76]—organoids comprise only ‘normal’ cells. For the purpose of this review, we provide specific definitions of glioma spheroids (GSs), glioma tumouroids (GTs) and brain organoids (BOs) (Table 2 and Figure 1). Each of these models can be cultured in liquid media, here referred to as ‘free’, or further developed by suspension in a 3D extra-cellular matrix, here referred to as ‘matrix-supported’, ‘spheroid’, ’tumouroid’ or ’organoid’, via the schema presented in Table 2 and depicted in Figure 1. This will aid in comparison between and within model types.

### 3.2. Free Spheroid and Tumouroid Models

Free 3D culture typically involves an aggregate of cells formed in a low-attachment dish/well [11,77] (as demonstrated in Figure 1), microfluidic chamber [78], or by suspension either by magnetic levitation [79], rotary cell culture system (RCCS) [73], or by hanging droplet.

GSs are formed from immortalised, or tumour-derived HGG cell lines under conditions which do not select for stem cell renewal. This contrasts with GTs, which are formed by culturing cells in media that promotes the maintenance of CSCs. GTs are also distinguished by the maintenance of increased cellular heterogeneity, which reflects the mixed cell populations that characterise tumours in-vivo [80].

Patient-specific therapy response [25] and retention of stemness [12,33,76,81] in pGBM and gene expression profiles of adult GBM tumours are evidenced in free GTs [2]. GTs have been used to demonstrate the role of HDAC6 in temozolomide (TMZ) resistance [82], investigate hedgehog signaling [83] and identify aurora A kinase (AUKRA) as a potential drug target in pGBM [69,72]. Please see Table 3 for a list of methods for the formation of free tumouroids or spheroids.

### 3.3. Matrix-Supported Spheroid and Tumouroid Models

Matrix-supported 3D culture involves the use of either naturally occurring extracellular matrix extracts or synthetic scaffolds [86,87]. These act as porous support structures typically containing components of synthetic or animal-derived ECM [88]. Cell suspensions [1] or pre-formed GSs or GTs can be added to the gel precursors before crosslinking, which once polymerised, support the cells from all sides. Encapsulating matrices allow cells to produce their own native 3D ECM [44], which is advantageous for brain tissue culture [38,45,47,66] and results in a gene expression profile closer to the parent tissue [2]. Pseudo-3D culture can also be performed by culturing cells on the surface of 3D hydrogels and immersed in liquid media [66].

Matrix-supported models have been developed using collagen I, laminin, gelatine, fibrin, HA and combinations thereof, the formulations of which have been extensively reviewed elsewhere [44,88]. Spheroids encapsulated in hydrogel droplets have been used to demonstrate the formation of central hypoxia [32] as well as providing a compact medium into which cells can migrate [1,15].

Since HA is present in high concentrations in brain ECM, and its stiffness can be tuned [38], HA-based hydrogels are particularly suited to supporting the culture of GBM. A higher concentration of HA was shown to induce resistance to erlotinib [47], TMZ [89] and dasatinib [89] in dissociated GTs [89] and single cells [47] within a HA-based gel and increased motility in cells cultured on the surface of gels [45]. Increased adhesion-mediated invasiveness was also observed [47].

Matrix-supported models are critical for analysing invasion, as they provide a replicate of the stroma surrounding tumours through which cancer cells must navigate in order to metastasise. Zhang et al. demonstrated this with spheroids and tumouroids formed from GBM cell line U251N and primary tumour specimens, and compared the migration through a collagen-based gel when treated with an experimental chemotherapy compound [1]. Migration of cell sheets was also observed from the periphery of GSs encapsulated in reconstituted basement membrane (rBM)-based hydrogels containing HA, with a speed dependent on HA concentration. Furthermore, sheet-like migration was also observed in GTs implanted in rat brains, supporting the validity of the matrix-supported spheroid model [15].

Concerning pGBM, although there are over 60 pGBM cell lines in existence [90], there were no pGBM matrix-supported models available on PubMed at the time of writing. Table 4 contains details of studies utilising matrix-supported adult GBM models.

Regarding molecular considerations, a microfluidic matrix-supported tumouroid model has been used to demonstrate that Interstitial fluid flow (IFF)-induced increases in invasion can be neutralised by blocking a patient-specific combination of CXCR4, CXCL12, and CD44 [70].

These findings suggest that hydrogel cultures that incorporate aspects of brain mechanics and ECM can faithfully recapitulate relevant HGG behaviours. However, these models do not recapitulate the diverse interactions between the cancer cells and the cell-laden TME. This could be achieved in part by incorporating a model of the healthy brain in the form of a cerebral organoid.

## 4. In-Vitro Models of Healthy Brain

Organoid cultures are organ-like 3D models generated from hESCs, induced pluripotent stem cells (iPSCs) or adult stem cells (ASCs) [91,92,93]. Organoids are defined as a self-organised 3D structures comprising multiple cell types with organisation, organ-like function and gene expression indicative of the organ being modelled [94]. Healthy BOs can model healthy brain elements for studying initiation, progression and invasion of HGG.

Table 5 presents a list of organoid models used for GBM research. The formation of organoids is heavily influenced by the microenvironment [23], which encompasses the provision of growth factors, morphogens, cell–cell interactions and cell-matrix interactions. These elicit changes in cell migration, differentiation and proliferation [23]. BOs have been generated from primary ESC and hESC cell lines [11,75], or iPSCs by exposing them to brain-mimetic stiffness cues [75,91,95] or specialised medium regimens [11,91]. BOs can be complex and are able to recapitulate the organisation and expression equivalent to a 20-week old human foetal brain [11]. Furthermore, organoids can display a primitive ventricular system, neural rosettes, microvasculature and expression of neural stem cell markers [11,96]. Although sophisticated, organoid culture has certain caveats such as an upper size limit imposed by unphysiological hypoxia. This has led to the use of shaking bioreactors [97] and millifluidic systems [98], for example, that improve media exchange to extend that limit [98].

## 5. Cerebral Organoid/Glioblastoma Co-Culture

Co-culture of primary GTs with BOs may be the most sophisticated model available in terms of recapitulation of the cancer environment. See Table 6 for a list of GT/BO or GS/BO co-culture combinations reported in the literature for modelling GBM (no use of pGBM was found). For example, Nayernia et al. co-cultured human pluripotent stem cell (hPSC) line H1-derived cerebral organoids with patient-derived CSCs in neural induction media in low-attachment plates to form free co-cultures. They observed radial invasion of GBM cells into the organoid tissue and increases in ECM-related genes that had a high impact on patient mortality [77]. These complex co-cultures were also produced by co-locating organoids and patient-derived CSCs, where tumouroids formed and infiltrated the BOs [11,75]. The extent of infiltration was patient-specific and in some cases, 30% of the organoid volume was overtaken by GBM after three weeks [75]. Additionally, levels of patient-specific EGFR were retained [11,99], which is rapidly lost when cells are cultured in 2D on stiff tissue culture plastic [99,100].

Others have studied tumour initiation by generating cerebral organoids and inducing oncogenesis with a combination of p53 knockout and transduction with an oncogene (HRasG12V) [75]. This induced oncogenesis [96] and the resulting organoids recapitulated the lethality observed [75]. Furthermore, BO/GT models were discovered to support the formation of TMs [11], which could be key to modelling the most aggressive GBMs and testing related targets, such as GAP-43. When GAP-43 was knocked down in a mouse model of GBM, the therapy-resistant effect of TMs was neutralised [29,47] and invasion and proliferation decreased [47]. Tweeny-homolog 1 (Ttyh1) has also been implicated in the invasiveness of TMs at the tumour leading edge [30], so may also present a viable target. Together, these studies imply that BO/GT co-cultures could be valuable for generating patient-specific models. These co-culture models are complex and time consuming to produce. Modern technologies such as 3D bioprinting may be able to streamline these processes to producing complex co-culture models.

## 6. Bioprinted Organoid/Tumouroid Models of High-Grade Glioma

Bioprinting is a form of matrix-supported cell culture where a specialised 3D printer is used to deposit cell-laden bioinks onto a printing bed, the two most common approaches being extrusion (filament) or ink-jet (droplet) approaches [86]—see Figure 2 for a graphical representation of each. Ink-jet printing is a no-contact option where droplets of bioink are released from a nozzle by applying heat or electric current [101] and can even be done with a modified office printer [102]. Extrusion printing is done by applying pressure—either pneumatically or with a piston—to a reservoir of bioink, thereby forcing it through a nozzle. The nozzle can then be manipulated using a CAD file to produce specially-controlled patterns of bioink [101]. An alternative to these more common printer types, a LASER-based printing method called LASER-based direct writing, can manipulate single cells and place them individually [103]. The more specialised systems for bioprinting have been reviewed elsewhere [104,105].

Bioinks are hydrogels that contain living single cells or cell conglomerates (spheroids or tumouroids). Laying down gel constructs in an additive way using 3D printing offers an unprecedented level of control over the spatial distribution of various cell/gel combinations [2]. The first bioprinted cell/hydrogel constructs were fabricated with a gelatine/alginate/fibrin bioink laden with HeLa cells [106]. Bioinks laden with GBM cells have since been developed from collagen alone [107], or in combination with alginate [2], gelatine methacrylate (GelMA) [48,108] or HA [109,110]. Decellularised animal brain ECM can also be formed into an ink [2,110,111], as well as a host of synthetic polymers including polyethylene glycol (PEG) [112]. Table 7 presents a list of studies that utilise bioprinting of GBM-laden bioinks.

### 6.1. Addition of Stromal Components to 3D Bioprinted High-Grade Glioma Models

Of the environments in the tumour stroma, an advantageous one to model is the vasculature, through which therapies are delivered. Alternatively to bioprinting, a model of vasculature can be generated through self-organisation of stem cells [11], coating organoids in endothelial cell-laden Matrigel [113] and through passive angiogenesis in the host (mouse) [114].

Multiple cell types have been bioprinted to form more complex systems with characteristics indicative of brain [2,48,107,111]. Models of vasculature can also be incorporated [67,96,97,115]. Even more importantly for GBM research is a 3D model for the blood–brain barrier (BBB), which can be done using complex microfluidics [116]. Table 8 contains details of several microfluidic vessel models for GBM that represent this emerging trend. A microfluidic channel within a collagen/laminin hydrogel lined with human umbilical vascular endothelial cells (HUVECs) serves as a vessel model [107]. Primary GSs injected a distance from this microvessel model migrated towards the microfluidic vessel at an increasing speed with increasing laminin concentration within the encapsulating bioink [107]. Demonstrating the validity of this model, the total distance migrated and TMZ response were patient-specific. The same cells cultured in 2D failed to exhibit patient-specific responses [107].

Vasculature has also been generated in a healthy BOs by coating with a Matrigel/ HUVEC layer. The HUVECs infiltrated the organoid and developed vasculature [113]. Co-culture of GBM cells with monocytes [2], macrophages [2,111] and astrocytes [108] have also been reported in 3D bioprinted models in order to model a simplified immune response.

GS co-cultures can be bioprinted with high viability [48,111,117] and retention of stemness [117]. However, since bioprinting is fairly new, new drug targets have yet to be discovered using bioprinted models. They have, however, along with microfluidic systems, potential to be valuable drug screening tools for GBM [78,118,119]. While currently there are no reports of bioprinted pGBM, such models are likely to be equally valuable in the study of pGBM.

### 6.2. Future Trends in 3D Bioprinting

The 3D bioprinting community has been moving towards rapid creation of complex multi-cell and multi-gel constructs that model increasingly complex structures. Bioprinting a wider variety of cell types in an ever-increasing library of structures is driven by developments in the printing technology itself—by adding the ability to print multiple materials [115,120]—as well as the development of increasingly advanced biomaterials with synthetic as well as animal-derived constituents [121] to accommodate primary cells in appropriately tailored environments. Additionally, some groups are producing 4D biomaterials, so that the degradation of the crosslinked polymer can be controlled over time via light [122] or temperature [123], or the cells influenced by an electrically conductive polymer [124].

Additionally, alongside developments in microvessels-mimicking [115], there has also been recent development in cancer-on-a-chip systems [111,119,125]. While these microfluidic chips are not necessarily bioprinted, they offer various degrees of functionality afforded by the versatility of microfluidics. Advanced cancer-on-a-chip systems have been used to recapitulate response to chemoradiotherapy [111] and are especially suited to investigating mechanisms of response to fluid shear [70,126], for example. Three-dimensional GS-on-a-chip systems are already a reality and they exhibit hypoxia and related increases in stem cell markers [127].

## 7. Conclusions and Future Outlook

Unfortunately, even given these advances in methods and technology, there are a distinct lack of advanced models of pHGG compared to adult HGG. As discussed, tools are available for the production of high quality patient-specific HGG models on a time scale relevant for patients even with the most dire prognosis. Advanced bioengineered models of GBM are rapidly progressing and there is no doubt that they can make a real impact on the outcomes of patients. These models will be necessary to undertake high-throughput screening to test for potential new therapeutic agents.

In this review, comparisons have been made between various effective culture modalities used to incorporate important aspects of the TME into pre-clinical models of GBM and pGBM. As is clear, traditional methods are being upgraded to complex hybrid techniques that incorporate combinations of controlled extracellular matrix composition, mechanical properties, fluid properties, cell-stroma interactions and solute concentrations, among others. These are being effectively utilised to gain insight into the complex mechanisms behind GBM. Notably, the combination of bioprinting and microfluidics has great potential to facilitate the co-culture of tumour, stem and stromal cells in a controlled, dynamic TME that elicits behaviours increasingly true to the patient’s own tissues.

There are certainly still many unanswered questions about pHGG. Moreover, of the work published on PubMed since the first mention of pre-clinical models of GBM in 1969, the total number of those mentioning pGBM comprise only 4.5%; there is a need for researchers to first take up the use of available pHGG cell lines [90] so that progress can then be made in the areas of tumouroids, co-culture and complex 3D engineered models (bioprinted and microfluidic alike). There is substantial progress to be made in this area, and once it is, steps can be taken to make all HGGs less of a burden.

## Figures and Tables

**Figure 1 ijms-22-02962-f001:**
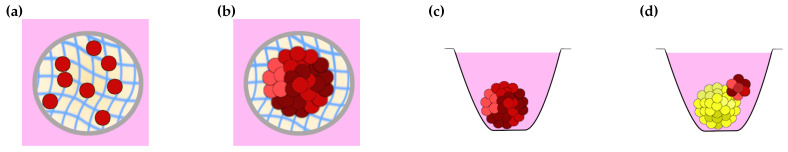
Schematics depicting various key culture modalities employed to model HGG. (**a**) matrix-supported cell suspension, (**b**) matrix-supported glioma spheroid (GS) or tumouroid (GT), (**c**) free spheroid and (**d**) free organoid/tumouroid (BO/GT) or organoid/spheroid (BO/GS) co-culture.

**Figure 2 ijms-22-02962-f002:**
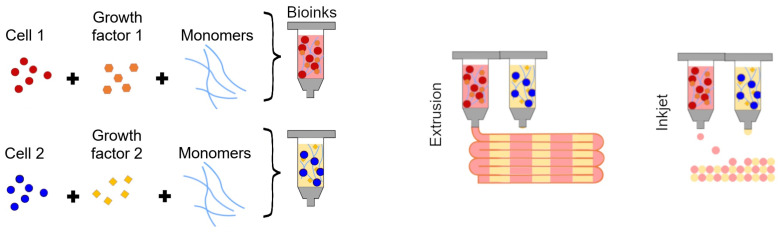
A schematic representation of co-printed disease models with two distinct gel formulations arranged spatially with filament- and droplet-style bioprinting.

**Table 2 ijms-22-02962-t002:** Definitions of the different types of GBM/brain models used throughout this review to compare and contrast existing literature. All models can be matrix-supported orfree.

Model Type	Definition
Glioma spheroid (GS) (with serum)	Dense conglomerate of cells cultured in serum—growth of CSCs not specifically promoted.
Glioma tumouroid (GT)	Tumour organoids generated by growing primary tumour material in suspension under defined media conditions in the absence of serum, CSCs specifically promoted and cellular heterogeneity maintained.
Brain organoid (BO)	Derived from stem cells under specific media and growth conditions to promote tissue lineage differentiation, displays some functionality and morphological features of model organ.
GS/BO and GT/BO	Glioma spheroid or tumouroid co-cultured with a BO.
Free	Single cells/spheroid/tumouroid suspended in liquid medium.
Matrix-supported	Single cells/spheroid/organoid encapsulated in a 3D matrix.

**Table 3 ijms-22-02962-t003:** Key methods for the formation of free tumouroids and the findings generated with theiruse.

Model Type	Cell Origin	Culture Method	Findings	Ref.
**Adult GBM**				
Free tumouroid	Cerebral organoid generated from hESC cell line H1.	Oncogenesis transduced with oncogene and knockdown of p53.	Tumouroids can be generated from cerebral organoids via gene manipulation.	[75]
Free tumouroid	Dissociated GBM specimens.	Suspended in serum-free media.	Tumouroids recapitulated the morphology and expression profile of parent GBM tumours.	[3]
Free tumouroid co-culture	GA-MSCs and CSCs were isolated from surgical specimens of GBM stroma and GBM, respectively.	Dissociated and resuspended in liquid differentiation media.	Stromal GA-MSCs excrete exosomes that increased proliferation of GSC xenografts and decreased median survival of the host animals when pre-treated with stromal GA-MSCs-derived exosomes.	[26]
Free tumouroid/ spheroid	Patient-derived GSCs/ cell line U87	Non-adherent plates.	All patient-derived tumouroids from primary GSCs were Nestin and Sox2 positive. Chemotherapeutics were effective only on 3D U87 spheroids. Tumouroids from the one recurrent cell line were the most drug-resistant. TMZ efficacy was patient-specific.	[84]
**Paediatric GBM**				
Ex-supported tumouroid (passaged in PDX models then extracted)	Specimens of pGBM	Xenografts of human pGBM patients with therapy-naive, recurrent and lethal disease were extracted, minced and enriched for CSCs.	An AUKRA inhibitor was most effective on therapy-naive tumouroids, followed by recurrent ex-xenografted tumouroids.	[69]
Free tumouroid	Tumour specimens from six pGBM patients	Stem cell population expanded via specialised media.	EGFR and PDGFRA amplification and deletion of RB1, CDKN2A/B & PTEN was observed.	[33]
Free tumouroid	Dissociated pGBM specimens from two patients	Suspended in serum-free media.	Stemness markers nestin, CD133, Sox2, melk, PSP and bmi-1 were expressed.	[85]
Free tumouroid	Dissociated pGBM specimens from 14 patients	Suspended in neural stem-cell media.	Stemness markers CD133 and Nestin were expressed and self-renewal was retained even when secondary tumouroids were formed from a single cell.	[81]

**Table 4 ijms-22-02962-t004:** Key methods for the formation of matrix-supported adult high-grade glioma spheroid and tumouroid models and the findings generated with their use. No matrix-supported models of paediatric HGG were found at the time of writing.

Model Type	Cell Origin	Culture Method	Findings	Ref.
Matrix -supported spheroid	GBM lines E98, E468 & U-251MG	Spheroids formed with hanging drop and implanted in nude rats, rat brain slices, rBM-based hydrogel layers or 3-layers of astrocytes. Hyaluronic acid was added to media.	Migration on brain slices was through blood vessels. Spheroids on rBM hydrogel and astrocyte layers recapitulated some migratory patterns seen in live rat brains. Higher HA concentration in media induced more rapid migration.	[15]
Matrix -supported spheroid	GBM cell line U251N	Hanging drop then embedded in collagen gel.	TMZ was effective in dose- and time-dependent manner	[1]
Matrix -supported spheroid	Patient-derived cell lines K301, GBM6, GS024 & GS025	Tumouroids were formed in suspension, dissociated, then transferred to HA-based hydrogel in a microfluidic chip.	Higher HA induced proliferation and drug resistance.	[89]
Matrix- supported tumouroid	Patient-derived CSCs.	Low-attachment plates and neurobasal media then encapsulation in HA/collagen hydrogel. Interstitial pressure was applied by deferentially filling a Millipore insert in a cell culture well.	Increased flow through the channel induced patient-specific increase in migration between 1.3 and 1.5-fold. With knockdown of CXR4, CXCL12 and CD44, a flow-induced increase in migration was neutralised.	[70]

**Table 5 ijms-22-02962-t005:** Methods for the formation of BOs and the findings generated with their use.

Model Type	Cell Origin	Culture Method	Findings	Ref
Brain organoid	hESC cell line H9	Differentiation media	Organoids were transduced to invoke oncogenesis. The number of modified, malignant cells surpassed healthy organoid cells over weeks.	[75]
Brain organoid	hESC cell lines H1, H6 or H9	Matrigel-coated plates & differentiation media	A primitive ventricular system and neural rosettes were formed & a proliferative zone of neural stem cells was present.	[11]
Brain organoid	iPSCs	Differentiation media & transfer to orbital shaker or millifluidic device	Millifluidic media exchange successfully reduced size of necrotic and hypoxic regions. No overall size difference was observed.	[98]
Brain organoid	hESCs	Low-attachment plates & differentiation media	Induction of common GBM genes with electroporation resulted in malignant cells overtaking healthy organoid cells within a month.	[96]

**Table 6 ijms-22-02962-t006:** Key methods for the formation of co-culture models from various combinations of tumouroid/spheroid/organoid and the findings generated with their use.

Cancerous Constituent	Culture Method	Healthy Brain Constituent	Culture Method	Findings	Ref.
Tumouroid	Dissociated primary CSCs cultured inlow-attachment plates with differentiation media	Brain organoid	hESC cell line H1 cultured inlow-attachment plates & differentiation media	Radial migration of tumouroid cells. Modification of ECM related expression similar to in-vivo.	[77]
Spheroid	SK2176 GBM cell-line cultured inlow-attachment plates	Brain organoid	hESC cell line H1 cultured in differentiation media	Spontaneous attachment and invasion of tumour cells into cerebral organoid. 30% of organoid volume was invaded after 24 days. Degree of invasiveness in model correlated with lethality of orthotopically xenografted tumouroids.	[75]
GSC cell line insuspension	Co-culture	Brain organoid	hESC cell line H1, H6 or H9 culture inMatrigel-coated plates with differentiation media	Co-cultures were more resistant to chemo-therapeutic agents and radiation versus 2D cultures. EGFR levels of parent tissue were recapitulated in 3D co-cultures and absent in 2D analogues.	[11]
Transfection of 18 GBM-like gene mutations/ amplifications	Oncogenesis of organoid via electroporation	Cerebral organoid	Generated from EBs with differentiation media	GBM can be initiated by selective gene manipulation. Increased invasiveness, higher expression of invasion-related genes and lower expression of tumour-inhibitive genes were observed in gene-altered cells.	[96]

**Table 7 ijms-22-02962-t007:** Key findings reported with the use of bioprinted glioblastoma models. No bioprinted paediatric GBM models were found at the time of writing.

Model Type	Cells Used	Gel Material and Organisation	Findings	Ref.
Bioprinted matrix -supported co-culture	GBM cell line U87MG, GSC lines G166, G144 & G7 monocyte cell line MM6	RGD-alginate + <250 mg/L HA or collagen I. Central tumouroid was printed then surrounded by astroma-like cell-laden gel construct.	Printed GBM cells remained viable (>90%) for months and CSCs retained stemness. Temozolomide IC50 doubled for printed spheroids compared to 2D co-cultures. GBM cells printed alongside fibroblasts were more resistant to TMZ.	[2]
Bioprinted matrix -supported co-culture	GBM cell line GL261 & macrophage cell line RAW 264.7	GelMA was used as both GBM and stroma-like bioink to create a GBM tumour model enclosed by amacrophage-laden gel construct.	Shear-thinning GelMA decreased printing-related cell death. Macrophages migrated towards GBM cells in co-culture and GBM cells had 15-fold increases inGBM-specific markers compared to 3D and 2D mono-culture.	[48]

**Table 8 ijms-22-02962-t008:** Engineered organoid models of GBM incorporating a vasculature model.

Model Type	Gel Material and Layout	Findings	Ref.
3D GBM- vascular niche with patient- derived CSCs co-cultured with HUVECs	A straight fluidic vascular channel was printed with collagen I and lined with HUVECs. CSCs were seeded adjacent to the microvessel.	At the highest concentration of laminin (100 µg/mL), CSCs migrated 1.5× further than inthegel containing 10 µg/mL of laminin.	[107]
GBM-on-a- chip with continuous cell line U-87 and patient- derived line co-cultured with HUVECs	A circular fluidic vascular channel was printed in collagen and abioink developed from decellularised porcine brain ECM. GBM-laden hydrogel was printed in the centre of a ring of collagen gel containing HUVECs. This was surrounded again by amicrochannel with an outer boundary printed in gas permeable silicone.	GBM cells grew in dense spheres with ananoxia-normoxia gradient and peripheral pseudopalisading cells. Cells in the intermediate region excreted factors leading to microvessel formation in the periphery. In porcine brain-derived gel, angiogenesis, proliferation and expression of pro-angiogenic genes and ECM remodelling proteins increased. All patient-derived cells in co-culture with HUVECs exhibited a dose-dependent response to TMZ but those on-chip recapitulated clinical therapy resistance, unlike the same cells cultured in 2D and 3D monoculture. Following multiple treatments, GBM cells extracted from patients with a longer survival exhibited decreased metabolic activity even after treatment ceased, whereas the metabolic activity increased after treatment ceased in the cells originating from patients with a shorter survival.	[111]

## Data Availability

Not applicable.

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
