# Peer review of "Advanced Spheroid, Tumouroid and 3D Bioprinted In-Vitro Models of Adult and Paediatric Glioblastoma"

_ijms, 2021, doi:10.3390/ijms22062962_

Round 1

Reviewer 1 Report

Dear Authors,

The article presents an interesting review of the literature on the state of interdisciplinary knowledge on the actual use of 3D bio-printing in medicine. Chapter 3.1 on Three-dimensional in-vitro models of glioblastoma is of particular interest.

It seems, however, that there are several issues that can be improved at work:

  • In Chapter 6, I propose to describe in more detail 3D printing technologies from biomaterials and existing machines 3D printers.
  • In Chapter 1, I also recommend mentioning other potential applications of 3D printing in medicine, e.g. using FDM and electrospinning technology (Electrospinning on 3D Printed Polymers for Mechanically Stabilized Filter Composites, DOI: 10.3390 / polym11122034,
  • In chapter 6 I recommend to add proto of already printed models in biopriting and e.g. from some CT, X-Ray ect.
  • In Chapter 7 I also propose to mention the future trends in bio-printing development and the challenges.

With these changes, it appears the article may be republished in such form.

Regards,

Reviewer

Reviewer 2 Report

The authors reviewed the currently available in-vitro models of high-grade gliomas (HGG), with special attention to the pediatric HGG, concluding that, compared to the models of adult HGG, more researches are necessary for the pediatric ones.

I really enjoyed reading this manuscript, and both the references and the figures clarify and support the whole work.

I have only a question: which is, according to the authors' findings, the most promising way to better develop and improve pediatric models of HGG for future researches (if there is one)?
